# A Transversal Approach Combining In Silico, In Vitro and In Vivo Models to Describe the Metabolism of the Receptor Interacting Protein 1 Kinase Inhibitor Sibiriline

**DOI:** 10.3390/pharmaceutics14122665

**Published:** 2022-11-30

**Authors:** Romain Pelletier, Thomas Gicquel, Mélanie Simoes Eugenio, Pierre-Jean Ferron, Isabelle Morel, Claire Delehouzé, Marie-Thérèse Dimanche-Boitrel, Morgane Rousselot, Brendan Le Daré

**Affiliations:** 1Institut NuMeCan (Nutrition, Metabolism and Cancer), INSERM, INRAE, CHU Rennes, PREVITOX Network, 35033 Rennes, France; 2Clinical and Forensic Toxicology Laboratory, Rennes University Hospital, Univ Rennes, 35033 Rennes, France; 3IRSET (Institut de Recherche en Santé, Environnement et Travail)—UMR_S 1085, Univ Rennes, Inserm, EHESP, 35000 Rennes, France; 4SeaBeLife SAS, Station Biologique, Place Georges Teissier, 29680 Roscoff, France; 5Pharmacy Department, Rennes University Hospital, 35033 Rennes, France

**Keywords:** molecular networking, Sibiriline, metabolism, LC-HRMS/MS

## Abstract

Sibiriline is a novel drug inhibiting receptor-interacting protein 1 kinase (RIPK1) and necroptosis, a regulated form of cell death involved in several disease models. In this study, we aimed to investigate the metabolic fate of sibiriline in a cross-sectional manner using an in silico prediction, coupled with in vitro and in vivo experiments. In silico predictions were performed using GLORYx and Biotransformer 3.0 freeware; in vitro incubation was performed on differentiated human HepaRG cells, and in vivo experiments including a pharmacokinetic study were performed on mice treated with sibiriline. HepaRG culture supernatants and mice plasma samples were analyzed with ultra-high-performance liquid chromatography, coupled with tandem mass spectrometry (LC-HRMS/MS). The molecular networking bioinformatics tool applied to LC-HRMS/MS data allowed us to visualize the sibiriline metabolism kinetics. Overall, 14 metabolites, mostly produced by Phase II transformations (glucuronidation and sulfation) were identified. These data provide initial reassurance regarding the toxicology of this new RIPK1 inhibitor, although further studies are required.

## 1. Introduction

Receptor-interacting protein 1 kinase (RIPK1) inhibitors are emerging as a new pharmacological class proposed for inflammatory, autoimmune, and neurodegenerative diseases such as rheumatoid arthritis, ulcerative colitis, psoriasis, Alzeihmer disease, amyotrophic lateral sclerosis, or multiple sclerosis [1,2,3,4]. Among them, Sibiriline appeared as a novel drug inhibiting necroptosis (a regulated form of cell death involved in several disease models). Sibiriline was shown to protect liver functions on a RIPK1-dependent concanavalin A-induced hepatitis, positioning it as a potential drug candidate for the treatment of immune-dependent hepatitis [5]. However, the therapeutic potential of Sibiriline will depend on its risk/benefit ratio and pharmacokinetic profile, including knowledge of its metabolism and potential drug-drug interactions. To date, nothing is known about the metabolism of Sibiriline, and its comprehensive exploration would require the use of cross-sectional analytical approaches.

In vitro, numerous models exist to study metabolism, including acellular models such as human microsomes and cell-based models such as the HepaRG line. Differentiated HepaRG cells (a bipotent cell line capable of differentiating into either cholangiocyte- or hepatocyte-like cells under specific culture conditions) have been described as a robust model to explore both toxicity and metabolism of xenobiotics [6,7,8,9]. In particular, this model expresses the majority of the drug processing genes including major CYPs [10], SULTs [11], and UGTs [11,12] and has long-term functional stability whereas the human hepatocytes in primary culture lose their differentiated phenotype and drug metabolism over time. Moreover, HepaRG cells have shown greater metabolite production compared with human liver microsomes motivating its growing use [13].

To complete the in vitro data, several in vivo models are also described and allow the addition of some pharmacokinetic notions to the metabolism study [14,15]. In particular, the rodent model is currently widely used to study the metabolism of xenobiotics for which we do not have human in vivo samples, as in the case of new psychoactive products [16,17]. While the quantitative aspect of rodent metabolism cannot be extrapolated to humans, the qualitative identification of metabolites is particularly relevant.

In order to reprocess complex analytical data collected in vitro and in vivo, innovative bioinformatic approaches have recently emerged. This is notably the case of molecular networking (MN) which allows the organization and representation of untargeted tandem mass spectrometry (MS/MS) data in a graphical form [18]. Each node represents an ion and its associated fragmentation spectrum, the links between the nodes indicating similarities between spectra. By allowing structural information to be propagated through the network and facilitating sample-to-sample comparison, the MN approach provides valuable insights into drug metabolism [19]. Consequently, a semi-quantitative visualization of molecule repartition in different matrix samples can be provided by a multi-matrix approach. MN has already proven its interest in drug metabolism analysis for in vitro, in vivo clinical, or forensic purposes [9,19,20,21,22,23,24].

Going further, the knowledge evolution of metabolism mechanisms has allowed the development of in silico metabolism prediction algorithms, which are a convenient, open access, time-saving, and inexpensive tool to broaden the metabolite search and confirm data found in vivo or in vitro [25]. Several approaches can be used in metabolism studies to produce in silico systems and include models based on (i) the quantitative structure-activity relationship (QSAR), which assumes that molecules with similar structures potentially have similar metabolic properties, or (ii) quantum mechanical calculations used to predict reactivity and/or (iii) docking of potential substrates into the active site of the enzyme [26,27,28,29,30].

In the present study, we aimed to investigate the metabolic fate of Sibiriline in a cross-sectional manner using an in silico prediction, coupled with in vitro and in vivo experiments.

## 2. Materials and Methods

### 2.1. In Silico Metabolite Prediction

Sibiriline putative metabolites were predicted using Sibiriline SMILES string (OC(C=C1)=CC=C1C2NC3=NC=CC=C3C2) through the free online software GLORYx [31] and Biotransformer 3.0 [32]. Briefly, GLORYx integrates machine learning-based site of metabolism prediction with reaction rule sets to predict and classify the putative structures of metabolites that could be formed by phase I and/or phase II metabolism. Biotransformer 3.0 is an open-access software tool that supports the rapid, accurate, and comprehensive prediction of the metabolism of small molecules in both mammals and environmental microorganisms. All metabolite structures were generated using ChemDraw software (PerkinElmer, Inc., Waltham, MA, USA).

### 2.2. Pharmacokinetic Study in Mice

Mice treatment. The in vivo pharmacokinetic study of Sibiriline was performed with male Swiss mice. A dose of 5 mg/kg Sibiriline was administered to 18 mice intraperitoneally (IP). After the times of 15 min, 30 min, 1 h, 2 h, 3 h, and 4 h, the mice were sacrificed. Blood was collected and plasma was separated by centrifugation. The samples were frozen and stored at −80 °C before processing.

Samples extraction. Each mouse plasma sample (400 µL) was mixed with 1 mL acetonitrile to precipitate the proteins and extract the compound. The samples were then vortexed for 5 min and placed in an ultrasonic bath for 1 min. The precipitated proteins were sedimented by centrifugation (15,000× *g*, 5 min at 16 °C) and the supernatants were transferred to a microplate for analysis by LC-MS/MS.

Analytical method for the quantitation of Sibiriline in mice. All samples are analysed with a validated Ultra-High Performance Liquid Chromatography method coupled to a Shimadzu LC-MS 8030 triple quadrupole. Solutions of a standard range, prepared with plasma from untreated mice, were analysed in the same series of injections.

LC separation was performed using a gradient on a C18 Phenomenex Kinetex column (50 × 2.1 mm, 2.6 µm). The mobile phase consisted of water 0.5% formic acid (solvent A) and acetonitrile (solvent B) delivered at a flow rate of 0.5 mL/min using the following stepwise gradient elution program: initial conditions of 95:5 (A:B) run maintained from 0 to 1.2 min, run from 5:95 (A:B) at 1.2 min maintained from 1.2 to 1.4 min, run from 95:5 (A:B) at 1.4 min, maintained from 1.4 to 2.8 min, Injections of 2 µL into the LC–HR-MS system were performed in a thermostated column at 40 °C. The tuning parameters of mass spectrometry (MS) were optimized by direct infusion of Sibiriline at the concentration of 10 µg/mL in a 1:1 acetonitrile/water mixture into the ionization probe in positive ion mode. The detection parameters were optimized to increase the sensitivity and signal stability by infusion of Sibiriline. Multiple reaction monitoring transitions (i.e., 211.0 → 118.0 *m/z* and 211.0 → 183.1 *m/z*) were used respectively for Sibiriline quantitation and confirmation. These analytical parameters provided satisfactory separation and peak shapes of Sibiriline in less than 3 min, and retention times (RTs) for Sibiriline was 0.86 min.

### 2.3. In Vitro and In Vivo Metabolism Studies

Material. William’s E medium (ref: 12551032) was purchased from Gibco (ThermoFischer Scientific, San Jose, CA, USA). Penicillin-streptomycin was obtained from Life Technologies (Grand Island, NY, USA). Fetal Bovine Serum (FBS) was purchased from Eurobio (Courtaboeuf, France) and from Hyclone GE Healthcare Life Sciences (Logan, UT, USA). Hydrocortisone hemisuccinate was purchased from Serb (Paris, France). Dimethyl sulfoxide (DMSO), formic acid, and insulin were obtained from Sigma-Aldrich (Saint Louis, MO, USA). Sibiriline was generously donated by SeaBeLife Biotech (Roscoff, France).

Cell culture and treatment. Progenitor HepaRG cells were cultured as already described [10]. Briefly, HepaRG cells were seeded at a density of 105 cells/well in 96-well plates and cultured for two weeks in culture medium (William’s E medium (1X) supplemented with 10% FBS, 50 U/mL penicillin, 50 μg/mL streptomycin, 5 µg/mL insulin, 2 mM glutamine, 50 μM sodium hydrocortisone hemisuccinate and 2% DMSO). Cells were then cultured for two more weeks in the same medium supplemented with 2% DMSO to induce cell differentiation into cholangiocyte- and hepatocyte-like cells [6]. The detection of Sibiriline and its metabolites was performed using this model as already described [9,33]. Differentiated HepaRG cells were incubated with 100 µL of Sibiriline (10 µM) during H0, H8, H24, or H48.

Mice treatment. The metabolism study was performed in male C57BL/6J mice (Janvier Labs, Le Genest Saint Isle, France). A dose of 5 mg/kg Sibiriline was administered to 10 mice intraperitoneally (IP). After the times 1 h and 3 h, the mice were sacrificed. Blood was collected and plasma was separated by centrifugation. The samples were frozen and stored at −80 °C before processing. Mice were housed in cages in a conventional animal facility with a 12 h dark-light cycle with controlled temperature between 19 °C and 20 °C. Experiments were done in compliance with French laws and the institution’s guidelines for animal welfare. Authors were authorized to conduct animal experimentation by “la Direction des Services Vétérinaires” (License M Samson #A3523840). The project was authorized by the “Comité Rennais d’Ethique en matière d’Experimentation Animal” [CREAA] and the license was given by the “Ministère de l’Enseignement Supérieur, de la Recherche et de l’Innovation”, #32246-2021061616397414).

Samples extraction. In vitro samples (200 µL) and in vivo plasma samples (100 µL) obtained from mice at the Rennes University Hospital were extracted as previously described [20,33]. Briefly, samples were supplemented with 500 µL of methanol containing internal standard (risperidone-D4) and then extracted with 300 μL of 0.1 M zinc sulfate solution. After supernatant evaporation, the residue was dissolved in 200 μL of LC-MS grade water and transferred into chromatographic vials for LC-HR-MS analysis.

LC-MS settings. Non-targeted screening LC-HRMS/MS method used for MN building was performed as already described [9,33]. Briefly, liquid chromatography was performed on an Accucore Phenyl Hexyl (100 × 2.1 mm, 2.6 μm) (Thermo Scientific, San Jose, CA, USA) at 40 °C using an elution gradient at a flow rate of 500 µL/min during 15 min with 10 µL as injection volume. An orbitrap mass spectrometer was operated in positive ESI mode and the acquisition range was 100–700 *m/z*. Ionic precursor selection was performed in a “data-dependent” mode of operation, where the 5 most intense ions from the previous scan were selected for fragmentation (Top N of 5).

MN generation. Spectral data allowed us to generate MN using a semi-quantitative approach. Data acquisition, processing (i.e., MS data conversion, preprocessing, MS1 annotation, and generation of molecular networks), visualization, and network analysis have already been described [19]. Briefly, raw data were converted to an open MS format (.mzXML) with ProteoWizard’s MSConvert module [34]. The mzXML files were then preprocessed (deconvolution, de-isotoping, alignment, gap-filling) with MZmine 2 software [35]. The single mgf output file was then loaded on the Global Natural Products Social networking (GNPS) web-based platform in order to generate the multi-matrix molecular network [18]. For the use of high-resolution data, the basic parameters were modified to *m/z* 0.02 for the mass tolerance of precursor and fragment ions used for MS/MS spectral library searching, and *m/z* 0.02 for the mass tolerance of fragment ions used for MN. The minimum cluster size was set to 1. In addition, links between nodes were created when the cosine score was greater than 0.70, and the minimum number of common fragment ions shared by two MS/MS spectra was 6. Links between two nodes were only kept in the network if each node was in the top 10 most similar nodes.

The molecular network was visualized using Cytoscape 3.8.0 software [36]. The nodes were annotated by comparison with reference standards by spectral matching with the curated GNPS, mzCloud online mass spectral libraries, and information propagation [37].

Figure 1 shows the overall methodology of the cross-sectional metabolic study approach used in this study.

## 3. Results

### 3.1. In Silico Prediction of Sibiriline Metabolism

In silico prediction of Sibiriline metabolism was performed on GloryX and Biotransformer 3.0 freeware and results are shown in Table 1. Predicted Sibiriline derivatives with an assigned GloryX score were considered probable (P1 to P14), while derivatives without an assigned score were considered minority and less likely (P15 to P21). Overall, Phase I and Phase II metabolites were predicted. Major Phase I metabolites included hydroxylated (P2, P4, P5, P7, P13, P14, P16, P17, P18, P19, P20) and oxidated (P10, P12, P15) derivatives. Major Phase II metabolites included glucuronoconjugated (P1, P6, P9, P16, P17, P18, P19), sulfoconjugated (P3, P20) and methylated (P8, P21) derivatives.

### 3.2. In Vitro Sibiriline Metabolism

To assess in vitro Sibiriline metabolite production, differentiated human HepaRG cells were incubated with Sibiriline for 24 h. Analysis of culture media by LC-HRMS/MS allowed us to generate a molecular network. Nodes are labelled with the exact protonated mass (*m/z*) and the links are labelled with the exact mass shift. Nodes were linked together in cluster according to their MS² spectral similarities (Figure 2).

In the sibiriline-containing cluster (Figure 2B), a specific color was assigned to each type of biotransformation reaction (glucuronidation in blue, sulfation in green, oxidation + sulfation in black, oxidation + glucuronidation in yellow, 2 × oxidation + 2 × glucuronidation in orange, methylation + sulfation reaction in pink and methylation + glucuroconjugaison in salmon). Sibiriline was associated with 18 structurally related molecules. Using information propagation, a node identified using a spectral library can be used as a starting point to identify another node in the same cluster The aim is to determine the structure of an unknown molecule using structural information from known neighboring nodes. Especially, spectrally related molecules may have mass shifts corresponding to well-known biotransformation reactions as shown in Table 1 [20].

Figure 2 shows two compounds with the same exact mass (*m/z* 387.116) but with two different retention times (RT: 4.8 and 3.7 min). These nodes are linked to Sibiriline (*m/z* 211.087) with a mass shift of +176.032, corresponding to glucuronidation reaction. From these nodes, we found three compounds with *m/z* 403.113 that could correspond to a hydroxylated derivative of Sibiriline glucuronide (RT: 4.1, 4.3, and 5 min) due to the mass delta of +15.994. We also found mass shifts that could correspond to sulfation (+79.956), giving rise to two *m/z* 291.043 compounds (RT: 5.2 and 4.4 min) (Figure 2) as well as mass shifts that could correspond to oxidation + sulfation (+ 95.951), giving rise to two *m/z* 307.038 compounds (RT: 5.2 and 4.4 min). Due to the mass shift of +28.031 that might correspond to a double methylation, the *m/z* 415.150 and *m/z* 319.075 nodes could be methylated derivatives of Sibiriline glucuronide and Sibiriline sulfate, respectively. Lastly, we identified a mass shift of 192.060 that could correspond to a hydroxylation + glucuronidation of *m/z* 403.113 compound (sibiriline + 2 × hydroxylation + 2 × glucuronidation). Structure and annotation proposals were made based on the *m/z* found during in silico prediction. Taken together, these results suggest that these Sibiriline spectrally related molecules are putative Sibiriline metabolites.

### 3.3. Sibiriline Pharmacokinetic Study in Mice

To determine how Sibiriline is metabolized in vivo, a pharmacokinetic study was performed by treating Swiss mice with an IP injection of sibiriline (5 mg/kg) (Figure 3). Sibiriline rapidly reached a peak plasma concentration within 15 min. The half-life was estimated at 21 min. The volume of distribution at 3247 mL/kg is indicative of a hydrophilic molecule (Figure 3).

### 3.4. In Vivo Sibiriline Metabolism

In order to strengthen the confidence level of the potential metabolites that we found in vitro, we performed an MN approach following LC-HRMS/MS on plasma samples from mice treated with Sibiriline (5 mg/kg IP) (Figure 4). As Sibiriline is rapidly metabolised in mice (half-life: 21 min), we tested for the presence of metabolites at 1 h and 3 h after the Sibiriline IP injection. Analysis of plasma samples at different times after Sibiriline treatment allowed us to generate a corresponding multi-matrix molecular network. A specific color was assigned to each time (H1 in salmon, H3 in red) so that we can visualize the fate of molecules structurally close to Sibiriline. The different colored areas in each node represent concentrations of the corresponding compound in each condition in a semi-quantitative manner.

Sibiriline-containing cluster visual analysis revealed that all putative metabolites found in the mice plasma samples were found in our previous in vitro experiment (Sibiriline glucuronide; Sibiriline sulfate, hydroxy-sibiriline glucuronide and hydroxy-sibiriline sulfate), except *m/z* 227.081 compounds which were not found in vitro. Due to the mass shift of +15.994, these compounds could be hydroxylated derivatives of Sibiriline, as predicted in silico. Interestingly, we found that almost all Sibiriline structurally related molecules are in the highest concentrations at 1 h of incubation, compared with 3 h of incubation. Only *m/z* 403.114 (RT: 4.3 min), which was found outside the sibiriline-containing cluster, showed higher concentrations at 3 h, compared with 1 h. These results show that Sibiriline and its structurally related compounds disappear rapidly over time.

Taken together, these results suggest that the Sibiriline structurally related compounds found in vitro and in vivo using molecular networking are putative metabolites. Also, we show that differentiated HepaRG is a powerful model in metabolism studies. Table 2 reports putative Sibiriline metabolites found in silico, in vitro and in vivo and provides annotations from M1 to M8, with the letters a, b, and c denoting isomers.

## 4. Discussion

In this study, we aimed to investigate the metabolic fate of Sibiriline in a cross-sectional manner using an in silico prediction, coupled with in vitro and in vivo experiments.

Overall, this cross-sectional approach allowed us to identify 8 Sibiriline putative metabolites in mice plasma (in vivo) and 12 putative metabolites in a human in vitro model (M1 to M8, Table 2). The higher number of metabolites found in vitro compared with in vivo could probably be due to the accumulation of metabolites in the culture medium (unlike the mouse model which includes all pharmacokinetic steps). As we have already shown that the metabolites identified in the HepaRG supernatant were quite similar to those found in urine, a comparison with mouse urine would be interesting in this context [33]. Interestingly, the in silico predictions proposed the main compounds found in vitro/in vivo showing that this approach is suitable for Sibiriline and strengthening the confidence of our in vitro/in vivo results. Biotransformer 3 outperformed GLORYx in predicting Sibiriline metabolites based on in vivo/in vitro data. However, two inconsistent metabolites (*m/z* 109.029 and *m/z* 119.061) were predicted by the in silico approach, leading to caution in the interpretation of the results. Most significantly, these complementary approaches allow (i) to come closer to the comprehensiveness of a metabolic study and (ii) to increase the degree of confidence in the separately generated results.

In terms of biotransformation reactions, we showed in silico, in vivo, or in vitro that the metabolism of Sibiriline is mainly performed by phase II enzymes (*m/z* 291.044, *m/z* 307.038, *m/z* 319.075, *m/z* 387.119, *m/z* 403.113, *m/z* 415.149, *m/z* 595.183), despite phase I metabolites were also identified (*m/z* 227.082). These results constitute a first reassuring basis from a toxicology point of view, as phase II metabolism is widely regarded as detoxifying. These reactive metabolites can exert initial cellular stress, through many mechanisms, such as glutathione depletion, or binding to proteins, nucleic acid lipids, and other cellular structures. Toxicological studies on sibiriline will be interesting in this context.

Phase II metabolism produces very hydrophilic compounds and may explain the rapid elimination of conjugated compounds shown in vivo in this study. While commonly used laboratory animals such as mice have many anatomical, physiological, and biochemical similarities to humans, there are marked differences between the species in their ability to process a drug [38,39,40,41]. Therefore, the interpretation of the pharmacokinetic data in mice presented in this study must be weighed against these considerations and a pharmacokinetic study in humans would be interesting here to evaluate Sibiriline elimination. Also, future pharmacokinetic studies of repeated-dose Sibiriline should take into account the selected animal species because the pharmacokinetics of the molecules may differ between species [42]. In terms of metabolism and toxicity, further studies are also needed, due to the fact that a molecule may be toxic due to a particular metabolism in one species, but not toxic to other species [43].

In addition, this would be of great importance in apprehending the administration modalities if used for therapeutic purposes. A short half-life should necessitate the use of strategies to prolong the effect of the molecule and could include the use of continuous intravenous infusion [44], the use of liposomal forms [45], intramuscular [46] or subcutaneous [47] administration, or the use of sustained-release oral forms [48]. Also, the pharmacokinetic and pharmacodynamic parameters of each of these strategies should be able to be based on trough concentration or area under the curve to further optimize the administration modalities [49].

This study presents several limitations. First, pharmacokinetic data in mice should be interpreted with caution due to interspecies differences in drug processing as drug metabolism is generally faster in rodents than in humans. Also, the pharmacokinetic study was performed by IP which is not the route of administration in acute liver failure. Second, the analysis of metabolites in mouse plasma does not necessarily allow the visualisation of metabolites excreted in urine, and an analysis in this matrix could be of interest. Third, since we analysed the culture supernatant of differentiated HepaRGs in vitro, it cannot be excluded that reactive metabolites have conjugated to intracellular proteins, not allowing their detection. Lastly, since the data presented in this study are mostly qualitative, it is difficult to determine which are the main metabolites.

## 5. Conclusions

This cross-sectional approach to studying metabolism using in silico, in vitro, and in vivo models brings us closer to a comprehensive exploration of xenobiotic metabolism and is a valuable tool in the study of new drug candidates. In addition, we applied an original MN-based approach to make the best use of the complex analytical data acquired by LC-HRMS/MS. In this study, we propose 14 putative metabolites that have mainly undergone phase II biotransformations. Pharmacokinetic data in mice showed rapid elimination of Sibiriline and its metabolites. Overall, these data provide initial reassurance regarding the toxicology of this new RIPK-1 inhibitor, although further studies are required.

## Figures and Tables

**Figure 1 pharmaceutics-14-02665-f001:**
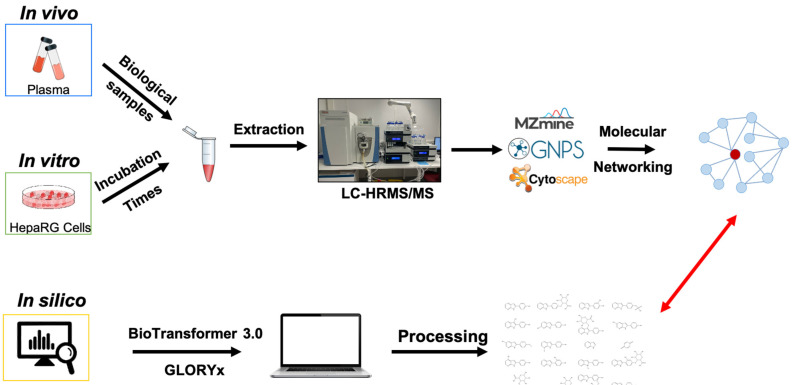
Overview methodology for cross-sectional study of sibiriline metabolism used in this study. LC-HRMS/MS = Liquid Chromatography coupled to a High-Resolution Tandem Mass Spectrometer.

**Figure 2 pharmaceutics-14-02665-f002:**
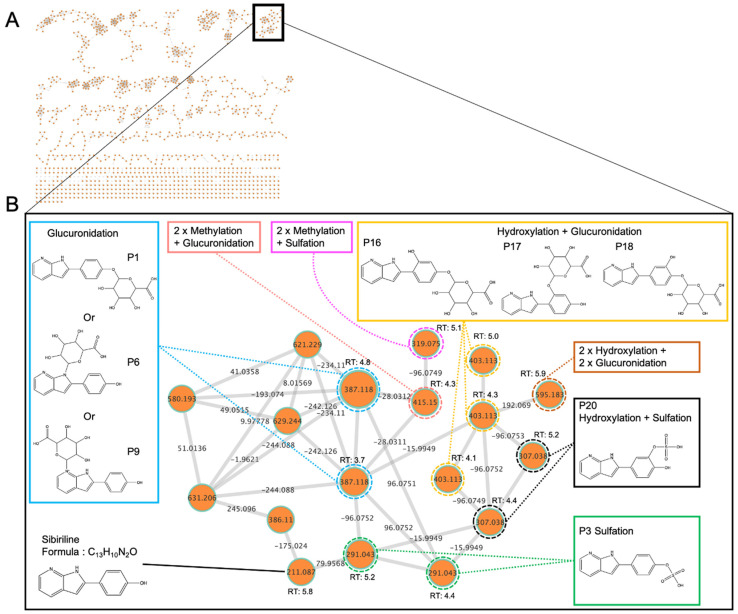
In vitro Sibiriline metabolism. Differentiated HepaRG were incubated with Sibiriline 10 µM during 24 h and culture supernatants were analyzed using LC-HR/MSMS. (**A**) The overall molecular network. (**B**) In this sibiriline-containing cluster, each type of biotransformation reaction was depicted in different colors (glucuronidation in bleu, sulfation in green, oxidation + sulfation in black, oxidation + glucuronidation in yellow, 2 × oxidation + 2 × glucuronidation in orange, methylation + sulfation reaction in pink and methylation + glucuroconjugaison in salmon). Nodes are labelled with the exact protonated mass (*m/z*), retention time (RT) in minute and the links are labelled with the exact mass shift.

**Figure 3 pharmaceutics-14-02665-f003:**
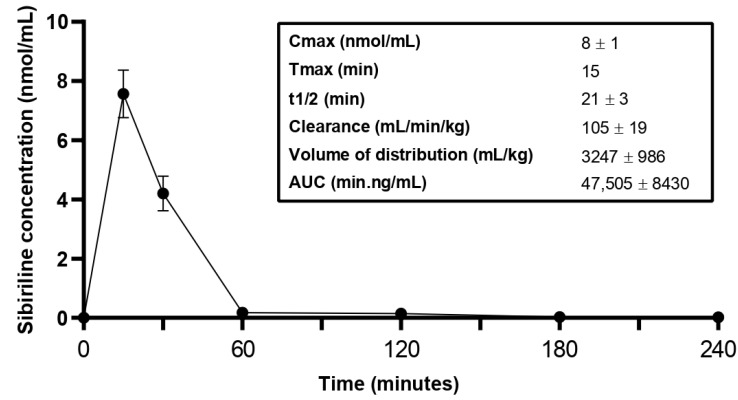
In vivo pharmacokinetics study of sibiriline. Evolution of the plasma concentration of sibiriline after intraperitoneal administration of sibiriline (5 mg/kg) in Swiss mice.

**Figure 4 pharmaceutics-14-02665-f004:**
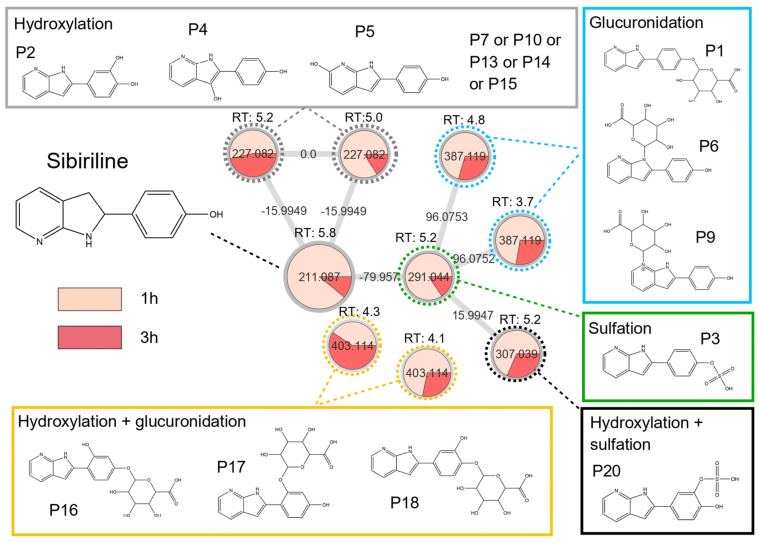
In vivo Sibiriline metabolism kinetic. Mice plasma sample collected 1 h or 3 h after intraperitoneal injection of Sibiriline (5 mg/kg) were analyzed using LC-HRMS/MS. In this sibiriline-containing cluster, each type of biotransformation reaction was depicted in different colors (glucuronidation in bleu, sulfation in green, oxidation + glucuronidation in yellow, and oxidation/hydroxylation in grey). Nodes are labelled with the exact protonated mass (*m/z*), putative chemical structures and the links are labelled with the exact mass shift. RT = retention time in minute.

**Table 1 pharmaceutics-14-02665-t001:** In silico predicted sibiriline metabolites and associated *m/z*, molecular structure, elemental composition, metabolic reaction, exact mass shift, predictive score (GLORYx software) and enzyme predicted to be involved in biotransformation (BT3: Biotransformer 3.0).

Predicted Metabolite (*m/z*)Elemental CompositionMetabolite Reaction	Putative Biotransformation ReactionExact Mass Shift	Molecular Structure	Prediction Software	Score and/or Enzyme Involved
Sibiriline (*m/z* 211.0871)C_13_H_10_N_2_ONone	Not applicable	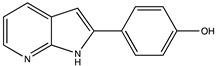	Not applicable	Not applicable
P1 (*m/z* 387.1192)C_19_H_18_N_2_O_7_O-glucuronidation	+C_6_H_8_O_6_+176.032	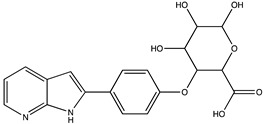	GLORYxBT3	0.948Predicted (UDP-glucuronosyltransferase)
P2 (*m/z* 227.0820)C_13_H_10_N_2_O_2_Aromatic hydroxylation	+O+15.994	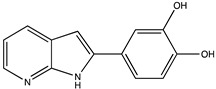	GLORYxBT3	0.656Predicted (CYP1A2)
P3 (*m/z* 291.0439)C_13_H_10_N_2_O_4_SSulfation	+SO_3_+79.956	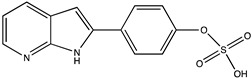	GLORYxBT3	0.628Predicted (Aryl sulfotransferase)
P4 (*m/z* 227.0820)C_13_H_10_N_2_O_2_Aromatic hydroxylation	+O+15.994	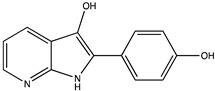	GLORYxBT3	0.348Not predicted
P5 (*m/z* 227.0820)C_13_H_10_N_2_O_2_Aromatic hydroxylation (para to carbon)	+O+15.994	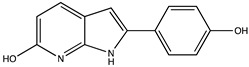	GLORYxBT3	0.348Not predicted
P6 (*m/z* 387.1192)C_19_H_18_N_2_O_7_N-glucuronidation	+C_6_H_8_O_6_+176.032	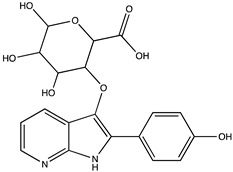	GLORYxBT3	0.308Not predicted
P7 (*m/z* 227.0820)C_13_H_10_N_2_O_2_Aromatic hydroxylation (para to nitrogen)	+O+15.994	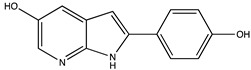	GLORYxBT3	0.295Predicted (CYP2C9)
P8 (*m/z* 225.1027)C_14_H_12_N_2_OMethylation (aromatic OH)	+CH_2_+14.015	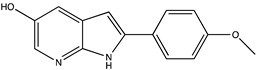	GLORYxBT3	0.28Not predicted
P9 (*m/z* 387.1192)C_19_H_19_N_2_O_7_+N-glucuronidation	+C_6_H_8_O_6_+176.032	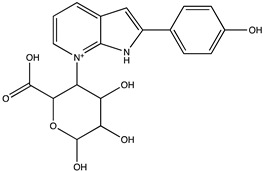	GLORYxBT3	0.224Not predicted
P10 (*m/z* 227.0820)C_13_H_10_N_2_O_2_N-oxidation	+O+15.994	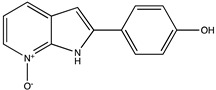	GLORYxBT3	0.212Predicted (CYP2C8)
P11 (*m/z* 119.0609)C_7_H_6_N_2_	Unknown	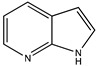	GLORYxBT3	0.1312Not predicted
P12 (*m/z* 109.0289)C_6_H_4_O_2_Oxidation of 4-substituted phenol to quinone	Unknown	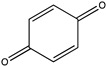	GLORYxBT3	0.1312Not predicted
P13 (*m/z* 227.0820)C_13_H_10_N_2_O_2_Aromatic hydroxylation	+O+15.994	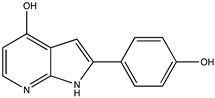	GLORYxBT3	0.072Predicted (CYP2C9)
P14 (*m/z* 227.0820)C_13_H_10_N_2_O_2_Aromatic hydroxylation	+O+15.994	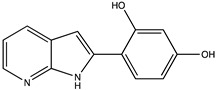	GLORYxBT3	0.028Predicted (CYP1A2)
P15 (*m/z* 227.0820)C_13_H_10_N_2_O_2_Oxidation	+O+15.994	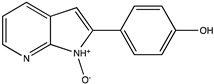	GLORYxBT3	NonePredicted (CYP2E1)
P16 (*m/z* 403.1141)C_19_H_18_N_2_O_8_Hydroxylation + O-glucuronidation	C_6_H_8_O_7_+192.027	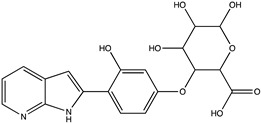	GLORYxBT3	NonePredicted (UDP-glucuronosyltransferase)
P17 (*m/z* 403.1141)C_19_H_18_N_2_O_8_Hydroxylation + O-glucuronidation	+C_6_H_8_O_7_+192.027	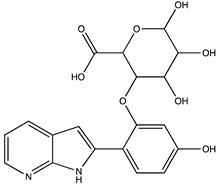	GLORYxBT3	NonePredicted (UDP-glucuronosyltransferase)
P18 (*m/z* 403.1141)C_19_H_18_N_2_O_8_Hydroxylation + O-glucuronidation	+C_6_H_8_O_7_+192.027	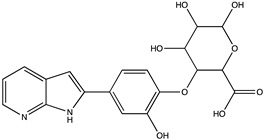	GLORYxBT3	NonePredicted (UDP-glucuronosyltransferase)
P19 (*m/z* 403.1141)C_19_H_18_N_2_O_8_Hydroxylation + O-glucuronidation	+C_6_H_8_O_7_+192.027	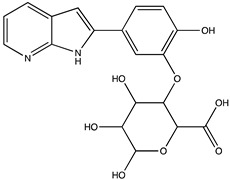	GLORYxBT3	NonePredicted (UDP-glucuronosyltransferase)
P20 (*m/z* 307.0388)C_13_H_10_N_2_O_5_SHydroxylation + O-sulfation	+SO_4_+95.951	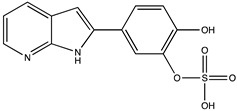	GLORYxBT3	NonePredicted (Aryl sulfotransferase)
P21 (*m/z* 241.0977)C_14_H_12_N_2_O_2_O-methylation	+CH_2_+14.015	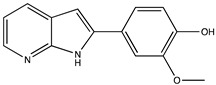	GLORYxBT3	NonePredicted (Catechol O-methyltransferase)

**Table 2 pharmaceutics-14-02665-t002:** Putative identified compounds or metabolites contained in the Sibiriline molecular network. RT: retention time; BT: Biotransformer 3; X = presence; - = absence.

Metabolite Proposal	Formula Proposal	Biotransformation	*m/z*	Mass Shift	RT (min)	In Vivo	In Vitro	In Silico	
								GLORYx	BT3
-	C_6_H_4_O_2_	Unknown	109.029	−102.058	-	-	-	X	-
-	C_7_H_6_N_2_	Unknown	119.061	−92.026	-	-	-	X	-
-	C_13_H_10_N_2_O	Parent molecule (sibiriline)	211.087	0	5.8	X	X	X	X
M1a	C_13_H_10_N_2_O_2_	Hydroxylation	227.082	15.995	5.0	X	-	X	X
M1b					5.2	X	-		
M2a	C_13_H_10_N_2_O_4_S	Sulfation	291.044	79.957	4.4	-	X	X	X
M2b					5.2	X	X		
M3a	C_13_H_10_N_2_O_5_S	Hydroxylation + sulfation	307.038	95.952	4.4	-	X	-	X
M3b					5.2	X	X		
M4	C_15_H_14_N_2_O_4_S	2 × methylation + sulfation	319.075	107.988	5.1	-	X	-	-
M5a	C_19_H_18_N_2_O_7_	Glucuronidation	387.119	176.033	3.7	X	X	X	X
M5b					4.8	X	X		
M6a	C_19_H_18_N_2_O_8_	Hydroxylation + glucuronidation	403.113	192.027	4.1	X	X	-	X
M6b					4.3	X	X		
M6c					5.0		X		
M7	C_21_H_22_N_2_O_7_	2 × methylation + glucuronidation	415.150	204.063	4.3	-	X	-	-
M8	C_25_H_26_N_2_O_15_	2 × hydroxylation + 2 × glucuronidation	595.183	384.096	5.9	-	X	-	-

## Data Availability

Not applicable.

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
