# Peer review of "A Transversal Approach Combining In Silico, In Vitro and In Vivo Models to Describe the Metabolism of the Receptor Interacting Protein 1 Kinase Inhibitor Sibiriline"

_pharmaceutics, 2022, doi:10.3390/pharmaceutics14122665_

Round 1

Reviewer 1 Report

This paper presents novel research on considering in vivo pharmacokinetic analysis with in vitro predictions to examine metabolism of a drug of high therapeutic value. This research seems to be preliminary and I anticipate that it will be followed by more work including a repeated administration in vivo study with a more relevant route of administration, such as oral or i.v. instead of i.p. on other species, such as rats, with sampling from not only plasma, but also urine and tissues, such as liver. It would be useful from a practical standpoint to see how those in vivo pharmacokinetic results compare with the in vitro metabolic predictions. 

The authors have described the objectives, methods, results and implications of their current work well. The authors have also acknowledged the limitations or data gaps left unaddressed by this work in the dicussion section. However, a more detailed discussion on future studies on repeated oral or i.v. administration on multi species with a case example or citations from literature of another drug that can demonstrate how in vivo results and in vitro predictions of studies conducted with alternate routes and in other species would be helpful. Also, a discussion on the possibility of expansion of the half-life of this drug could be discussed, which relate to recommendations on dosing, route, formulation etc. would benefit the discussion. 

Author Response

This paper presents novel research on considering in vivo pharmacokinetic analysis with in vitro predictions to examine metabolism of a drug of high therapeutic value. This research seems to be preliminary and I anticipate that it will be followed by more work including a repeated administration in vivo study with a more relevant route of administration, such as oral or i.v. instead of i.p. on other species, such as rats, with sampling from not only plasma, but also urine and tissues, such as liver. It would be useful from a practical standpoint to see how those in vivo pharmacokinetic results compare with the in vitro metabolic predictions.

The authors have described the objectives, methods, results and implications of their current work well. The authors have also acknowledged the limitations or data gaps left unaddressed by this work in the discussion section.

However, a more detailed discussion on future studies on repeated oral or i.v. administration on multi species with a case example or citations from literature of another drug that can demonstrate how in vivo results and in vitro predictions of studies conducted with alternate routes and in other species would be helpful. Also, a discussion on the possibility of expansion of the half-life of this drug could be discussed, which relate to recommendations on dosing, route, formulation etc. would benefit the discussion.

Response to reviewer:

Thank you for your comments. We thus completed our discussion page 15 of the manuscript:

“Therefore, the interpretation of the pharmacokinetic data in mice presented in this study must be weighed against these considerations and a pharmacokinetic study in humans would be interesting here to evaluate sibiriline elimination. Also, future pharmacokinetic studies of repeated-dose sibiriline should take into account the selected animal species because the pharmacokinetics of the molecules may differ between species [46]. In terms of metabolism and toxicity, further studies are also needed, due to the fact that a molecule may be toxic due to a particular metabolism in one species, but not toxic to other species [47].

In addition, this would be of great importance in apprehending the administration modalities if used for therapeutic purposes. A short half-life should necessitate the use of strategies to prolong the effect of the molecule and could include the use of continuous intravenous infusion [48], the use of liposomal forms [49], intramuscular [50] or subcutaneous [51] administration, or the use of sustained release oral forms [52]. Also, the pharmacokinetic and pharmacodynamic parameters of each of these strategies should be able to be based on trough concentration or area under the curve to further optimize the administration modalities [53].”

References:

  1. Jeon, J.-H.; Lee, J.; Choi, M.-K.; Song, I.-S. Pharmacokinetics of Ginsenosides Following Repeated Oral Administration of Red Ginseng Extract Significantly Differ between Species of Experimental Animals. Arch. Pharm. Res. 2020, 43, 1335–1346, doi:10.1007/s12272-020-01289-0.
  2. Natsch, A.; Nordone, A.; Adamson, G.M.; Laue, H. A Species Specific Metabolism Leading to Male Rat Reprotoxicity of Cyclamen Aldehyde: In Vivo and in Vitro Evaluation. Food and Chemical Toxicology 2021, 153, 112243, doi:10.1016/j.fct.2021.112243.
  3. Arensdorff, L.; Boillat-Blanco, N.; Decosterd, L.; Buclin, T.; de Vallière, S. Adequate Plasma Drug Concentrations Suggest That Amoxicillin Can Be Administered by Continuous Infusion Using Elastomeric Pumps. Journal of Antimicrobial Chemotherapy 2017, 72, 2613–2615, doi:10.1093/jac/dkx178.
  4. Powell, J.S. Liposomal Approach towards the Development of a Longer-Acting Factor VIII. Haemophilia 2007, 13, 23–28, doi:10.1111/j.1365-2516.2007.01502.x.
  5. Pacchiarotti, I.; Tiihonen, J.; Kotzalidis, G.D.; Verdolini, N.; Murru, A.; Goikolea, J.M.; Valentí, M.; Aedo, A.; Vieta, E. Long-Acting Injectable Antipsychotics (LAIs) for Maintenance Treatment of Bipolar and Schizoaffective Disorders: A Systematic Review. European Neuropsychopharmacology 2019, 29, 457–470, doi:10.1016/j.euroneuro.2019.02.003.
  6. Hall, S.; Isaacs, D.; Clements, J.N. Pharmacokinetics and Clinical Implications of Semaglutide: A New Glucagon-Like Peptide (GLP)-1 Receptor Agonist. Clin Pharmacokinet 2018, 57, 1529–1538, doi:10.1007/s40262-018-0668-z.
  7. Bialer, M.; Friedman, M.; Dubrovsky, J. Relation between Absorption Half-Life Values of Four Novel Sustained-Release Dosage Forms of Valproic Acid in Dogs and Human. Biopharm. Drug Dispos. 1986, 7, 495–500, doi:10.1002/bdd.2510070510.
  8. Gunaydin, H.; Altman, M.D.; Ellis, J.M.; Fuller, P.; Johnson, S.A.; Lahue, B.; Lapointe, B. Strategy for Extending Half-Life in Drug Design and Its Significance. ACS Med. Chem. Lett. 2018, 9, 528–533, doi:10.1021/acsmedchemlett.8b00018.

Reviewer 2 Report

Summary:

Receptor-interacting protein 1 kinase (RIPK1) inhibitors are emerging as a new target for autoimmune diseases such as rheumatoid arthritis, ulcerative colitis, psoriasis, Alzeihmer disease & multiple sclerosis. Sibriline is a novel drug reported to have RIPK1 inhibitory properties. There are not many research manuscripts on Sibiriline. This study focuses on identifying the metabolism of sibiriline using LC-MS, GLORYx and Biotransformer 3.0. The authors report 14 metabolites produced via the Phase II transformation.  

Comments:

Overall, this is a clear and well-written manuscript. The introduction explains the need for conducting this research and provides sufficient information needed for the readers.  The authors have used in-vitro and in-vivo models to understand the metabolism of sibiriline. The methods and results are generally appropriate. The authors have mentioned that their project should be followed with further investigation, but this is a good start and the findings of this study will certainly help develop projects for other scientists. The conclusions are consistent with the evidence provided.  Overall, this is a good-quality manuscript.  

Author Response

Overall, this is a clear and well-written manuscript. The introduction explains the need for conducting this research and provides sufficient information needed for the readers.  The authors have used in-vitro and in-vivo models to understand the metabolism of sibiriline. The methods and results are generally appropriate. The authors have mentioned that their project should be followed with further investigation, but this is a good start and the findings of this study will certainly help develop projects for other scientists. The conclusions are consistent with the evidence provided.  Overall, this is a good-quality manuscript. 

Response to reviewer:

Thank for your comments. We are delighted that this work has interested you.

Reviewer 3 Report

This work mainly describes a transversal approach combining in silico, in vitro and in vivo models to describe the metabolism of the receptor interacting protein 1 kinase inhibitor sibiriline. These authors simulated the metabolites of sibiriline in silico, and also completed the pharmacokinetic and metabolic studies of sibiriline in mice. However, compared with the other research (J Biotechnol, 9(11), 1446-57. J R Soc Interface, 17(164), 20190801. FEBS J, 284(18), 3050-3068. ), this study has neither innovations and Comprehensive in analysis methods nor new research ideas about the toxicological study of sibiriline. It is hoped that the author will refine the innovation of the article,and the workload of this study is not enough .Following points should be addressed:

1.        Setting basis of pharmacokinetic blood sampling time point.

2.        The method of administration in metabolite analysis is not consistent with the actual application. Why is intraperitoneal injection still chosen? Will there be a significant difference in metabolite identification compared with oral administration or other methods of administration?

3.        Why are different species of mice selected for pharmacokinetic and metabolic studies?

4.        The correlation of in silico, in vitro and in vivo models can be described as a figure.

5.        Even if sibiriline is metabolized quickly in the body, when identifying and identifying metabolites, whether more metabolites will be found by adding the time point of blood collection, in order to further improve the metabolic research.

6.        In Part 2.3. In vitro and in vivo metabolism studies, the blood collection time point for metabolic study was set as 2h and 4h in ‘Mice treatment’. It was not consistent with 1h and 3h in figuer 3.

7.        The main analysis method of this paper, LC-HRMS/MS, was not reflected in any figuers.
